# A machine learning-based risk score for prediction of mechanical ventilation in children with dengue shock syndrome: A retrospective cohort study

**Nguyen Tat Thanh** [1,2]*, **Vo Thanh Luan**[1], **Do Chau Viet**[1], **Trinh Huu Tung**[1], **Vu Thien**[3]

**1** Department of Infectious Diseases, Children Hospital 2, Ho Chi Minh City, Vietnam, **2** TB Department, Woolcock Institute of Medical Research, Ho Chi Minh City, Vietnam, **3** National Institutes of Biomedical Innovation, AI Nutrition Project, Health and Nutrition (NIBIOHN), Ibaraki, Osaka, Japan

☯ These authors contributed equally to this work.

* thanhhonor@gmail.com

**Data Availability Statement:** All relevant data are within the manuscript and its Supporting Information files.

## Abstract

### Background

Patients with severe dengue who develop severe respiratory failure requiring mechanical ventilation (MV) support have significantly increased mortality rates. This study aimed to develop a robust machine learning-based risk score to predict the need for MV in children with dengue shock syndrome (DSS) who developed acute respiratory failure.

### Methods

This single-institution retrospective study was conducted at a tertiary pediatric hospital in Vietnam between 2013 and 2022. The primary outcome was severe respiratory failure requiring MV in the children with DSS. Key covariables were predetermined by the LASSO method, literature review, and clinical expertise, including age (< 5 years), female patients, early onset day of DSS ($\leq$ day 4), large cumulative fluid infusion, higher colloid-to-crystalloid fluid infusion ratio, severe bleeding, severe transaminitis, low platelet counts (< 20 x $10^9$/L), elevated hematocrit, and high vasoactive-inotropic score. These covariables were analyzed using supervised models, including Logistic Regression (LR), Random Forest (RF), Support Vector Machine (SVM), k-Nearest Neighbor (KNN), and eXtreme Gradient Boosting (XGBoost). Shapley Additive Explanations (SHAP) analysis was used to assess feature contribution.

### Results

A total of 1,278 patients were included, with a median patient age of 8.1 years (IQR: 5.4–10.7). Among them, 170 patients (13.3%) with DSS required mechanical ventilation. A significantly higher fatality rate was observed in the MV group than that in the non-MV group (22.4% vs. 0.1%). The RF and SVM models showed the highest model discrimination. The SHAP model explained the significant predictors. Internal validation of the predictive model

**Funding:** The author(s) received no specific funding for this work.

**Competing interests:** The authors have declared that no competing interests exist.

**Abbreviations:** DSS, Dengue shock syndrome; MV, Mechanical ventilation; PICU, Pediatric intensive care unit; ML, Machine learning; LASSO, Supervised models, the Least Absolute Shrinkage and Selection Operator; LR, Logistic Regression; RF, Random Forest; KNN, k-Nearest Neighbors; SVM, Support Vector Machine; XGBoost, eXtreme Gradient Boost; SHAP, Shapley Addictive Explanations.

showed high consistency between the predicted and observed data, with a good slope calibration in training (test) sets 1.0 (0.934), and a low Brier score of 0.04. Complete-case analysis was used to construct the risk score.

## Conclusions

We developed a robust machine learning-based risk score to estimate the need for MV in hospitalized children with DSS.

## Introduction

According to a recent report by the World Health Organization (WHO) in 2023, there has been an unprecedented increase in dengue cases across all WHO-defined regions [1]. Since the beginning of 2023, approximately 6.5 million dengue cases and over 7,300 dengue-related fatalities have been reported, marking a historic surge [1]. Critical complications associated with dengue have contributed to high fatality rates, varying from approximately 5% to over 20% [2–8]. Severe complications among hospitalized children with dengue include significant plasma leakage, dengue shock syndrome (DSS), severe bleeding, acute liver failure, and respiratory failure [2–8]. Most notably, extensive plasma leakage causes excessive fluid accumulation in pleural and abdominal cavities, resulting in mild-to-severe respiratory failure. Significantly higher mortality rates have been reported in severe dengue patients with acute respiratory failure and MV [8,9].

Our research group recently conducted a pediatric cohort study in which approximately a quarter of hospitalized children with DSS developed severe respiratory failure requiring MV [4]. Another adult dengue cohort in Taiwan showed that 35% of patients with severe dengue required MV support [10]. In addition, cumulative fluid infusion > 15% and obesity have been shown to be important predictors of severe respiratory failure and the need for MV in severe dengue cases [3,8]. Despite the high prevalence and associated mortality of mechanical ventilation in severe dengue, there remains a paucity of data on the prognostic factors for MV among children with DSS [8,9,11]. Therefore, identifying the determinants of MV in children with dengue shock syndrome upon admission to the pediatric intensive care unit (PICU) is crucial for this patient population.

Notably, previous studies predicting the risk of death in hospitalized children with severe dengue have been limited by small sample sizes and a lack of robust statistical methodologies to validate predictive models [3–5]. Although our recent study, involving a cohort of 492 children with DSS, aimed to predict fatality, it was constrained by the small number of patients with fatal outcomes (26 deaths, 5.3%). Clinical evidence and literature review indicate that severe respiratory failure requiring MV is a strong independent predictor of mortality among children hospitalized with severe dengue [4,8–11]. Therefore, we used the need for mechanical ventilation as a surrogate outcome for mortality in children with DSS.

Machine learning has recently gained prominence for its high performance in prediction and classification across various medical fields, particularly in tropical diseases [12–16]. This study aimed to identify the most significant predictors of MV and develop a robust machine learning-based risk-scoring system to predict the need for MV in hospitalized children with DSS. The identification of prognosticators can aid in optimizing management protocols, thereby reducing the need for MV support and improving survival outcomes in patients with DSS.

## Methods

### Ethics statement

This study originated from the primary research titled "Prognostic model predicting mortality among patients presenting with dengue shock syndrome at Children's Hospital No. 2, during 2013–2022," which was approved by the Scientific Committee and Institutional Review Board (IRB) of Children's Hospital No. 2, Ho Chi Minh City, Vietnam (IRB No. 893/QD-BVND2 signed on 06-June-2022). A secondary dataset from the primary study was considered to pose less than minimal risk to participants. The requirement for informed consent was waived, and participants' identities were de-identified to protect patient confidentiality.

### Data source, study setting and population

This study utilized a dataset from the primary study, titled "Prognostic model predicting mortality among patients presenting with dengue shock syndrome at Children's Hospital No.2, during 2013–2022", which was approved by the Scientific Committee and Institutional Review Board (IRB) of the Children's Hospital No.2, Ho Chi Minh City, Vietnam (IRB. No. 893/QD-BVND2 signed on 06-June-2022) [4]. The requirement for informed consent was waived, and participants' identities were anonymized to ensure patient confidentiality. The study was performed in accordance with the principles of Good Clinical Practice and ethical guidelines of the Declaration of Helsinki. This study is reported in accordance with the Transparent Reporting of a Multivariable Prediction Model for Individual Prognosis or Diagnosis (TRIPOD) guidelines (**S1 Checklist**).

This single-institution retrospective study, known as the Vietnam Dengue-Infected Study (VNDIS), was conducted at the Children's Hospital No. 2, a large tertiary pediatric hospital in Ho Chi Minh City, Vietnam. The hospital has a capacity of 1,400 beds and serves patients from central and southern Vietnam. The study period spanned from 2013 to 2022.

The primary aims of this study were to identify the determinants of severe respiratory failure requiring MV and develop a machine learning-based risk scoring system for children with DSS. The eligibility criteria were age < 18 years, laboratory-confirmed dengue infection, and the presence of dengue shock syndrome [2]. The exclusion criteria were a lack of serologically confirmed dengue infection and missing data for variables of interest ($\geq$ 50%). The indications for mechanical ventilation in children with severe DSS were based on the national guidelines for dengue diagnosis and treatment, as outlined in **S1 File**.

### Study definitions

According to the World Health Organization (WHO) dengue guidelines (2009), laboratory-confirmed dengue infection was defined by the presence of dengue-IgM antibodies or a positive non-structural 1 antigen test [2]. Dengue shock syndrome and severe bleeding were diagnosed according to the WHO dengue 2009 guidelines [2]. Severe transaminitis was defined as elevated aspartate aminotransferase (AST) and/or alanine aminotransferase (ALT) levels of $\geq$ 1,000 IU/L [2].

### Study outcome

The primary outcome of the study was the requirement for mechanical ventilation in children with dengue shock syndrome who developed acute respiratory failure during their PICU stay based on data collected upon PICU admission.

### Candidate variables

From the original dataset, we identified 1,278 eligible participants with 82 variables [4]. We extracted 28 variables of interest that had a missing rate of < 50% and were consistent with the

disease pathogenesis (S1 Table). These variables were considered the initial set of candidate variables and sorted into the following main categories:

**Demographic data.**   Patient's age, sex, presence of underlying diseases.

**Clinical data.**   Severity of dengue shock, time of onset of dengue shock, severe bleeding, platelet transfusion, neurological status (AVPU scale), respiratory rate, systolic and diastolic shock indices, vasoactive inotropic score (VIS), cumulative volume of fluid infused (from the referral hospital and during 24h of PICU admission), and colloid-to-crystalloid fluid infusion ratio.

**Laboratory data.**   Hematological parameters, liver enzymes, and organ dysfunction biomarkers.

The percentages of the missing values for these variables are presented (S1 Fig).

## Feature selection and data preprocessing

A predetermined set of clinical and laboratory covariables was determined using clinical knowledge and the Least Absolute Shrinkage and Selection Operator (LASSO) method.

**Handling missing data.**   Missing data patterns were analyzed using the naniar, rms, VIM, and dlookr packages in R. Visualizations such as missing data patterns, Pareto charts, and hierarchical clustering were utilized to comprehend and manage missing values. Variables with more than 50% missing data were excluded from the statistical analysis. Multiple imputations were performed using the missRanger package to handle the missing values [17].

**Data cleaning and transformation.**   Categorical variables were converted into factors. Binary variables, such as the MV outcome, were converted into a numeric (0/1) format. Outliers and irrelevant categories were removed where applicable.

## Statistical analysis

All the steps involved in developing the prediction model and risk score are detailed in Fig 1.

**Model selection.**   This study employed various supervised machine learning models, including Logistic Regression (LR), Random Forest (RF), Support Vector Machine (SVM), k-Nearest Neighbors (KNN), and eXtreme Gradient Boosting (XGBoost).

**Data splitting and preparation.**   The dataset was randomly split into training (70%) and testing (30%) sets using stratified sampling to ensure a representative distribution of the outcome variable. One-hot encoding was applied to categorical variables and normalization was performed for numerical variables.

**Model training and hyperparameter tuning.**   Ten-fold cross-validation was used to train the models, ensuring robustness, and preventing overfitting. Hyperparameter tuning was conducted using a grid search to optimize the model parameters. The best model for predicting severe respiratory failure requiring MV was selected based on performance metrics evaluated on the test set.

**Performance metrics.**   Model performance was evaluated using accuracy, sensitivity, specificity, precision, area under the receiver operating characteristic curve (AUC), and F1 score.

## Model explanation

The Shapley Additive Explanations (SHAP) method, based on the best performance supervised model, was employed to interpret the contribution of each feature to the model predictions. The SHAP values provided a unified measure for interpreting features of importance and their impact on the model's output.

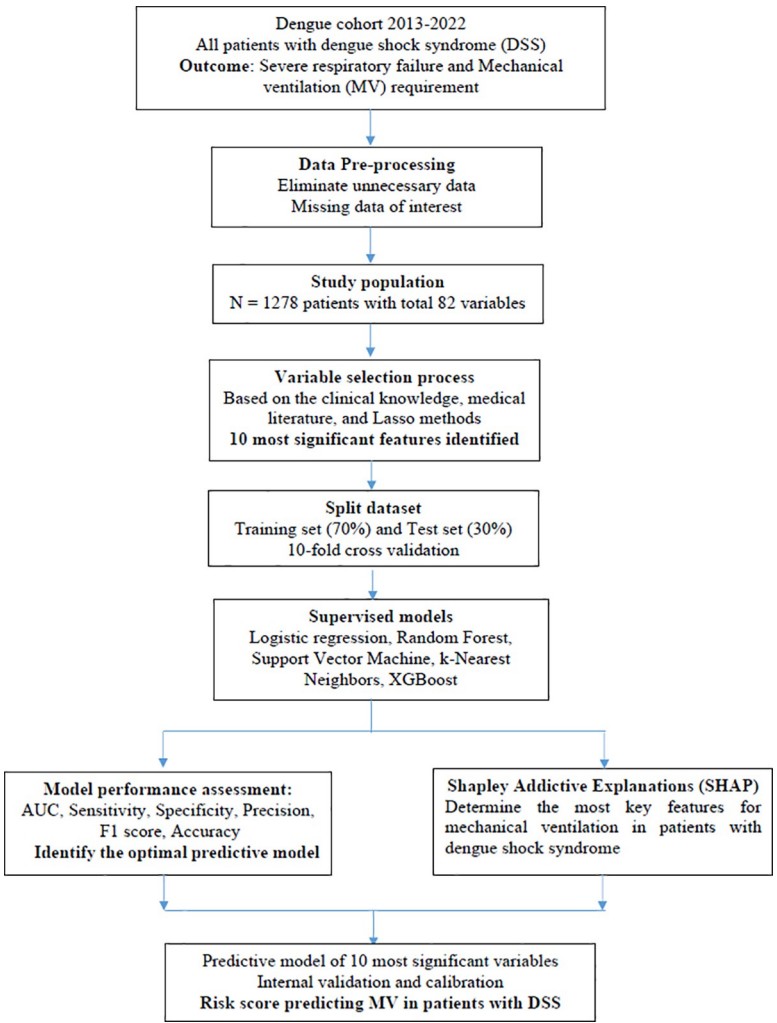

**Fig 1. The study flowchart for developing predictive models of mechanical ventilation in children with dengue shock syndrome.**

## Model validation and calibration, and risk scoring system

We utilized the Hmisc and rms packages in the R library to develop the LR model to predict the need for mechanical ventilation in pediatric patients with DSS. Variables included in the validation and calibration were selected based on the SHAP results and clinical expertise. These variables comprised age, sex, severe bleeding, low platelet counts, severe transaminitis, peak hematocrit, cumulative volume of fluid from the referral hospital and 24h of PICU admission, the colloid-crystal fluid infusion ratio, early onset day of DSS, and vasoactive inotropic score. The predictive model was internally validated using bootstrap sampling (n = 500) and its performance was assessed using a calibration plot to visualize the consistency between the predicted and observed values. A risk-scoring system (Dengue MV score) was developed for predicting mechanical ventilation in DSS children using the ten most important features identified from the best-performing supervised models based on complete-case data analysis. Cutoff values for the final covariables in the risk score were determined using our clinical knowledge and the existing medical literature. All statistical analyses were performed using R

(version 4.3.2) with packages, including rms, naniar, VIM, dlookr, missRanger, glmnet, caret, xgboost, SHAPforxgboost, and cutpointr. Statistical significance was set at $P < 0.05$.

## Results

### Baseline characteristics of study participants upon PICU admission

Between 2013 and 2022, approximately 2,000 children with DSS were admitted to the PICU, of whom 1,278 met the eligibility criteria and were included in the analysis. Of these, 170 patients (13.3%) experienced severe respiratory failure requiring MV, while the remaining 1,108 patients (86.7%) did not require MV. The baseline clinical and laboratory characteristics of the study participants are presented in Table 1. Patients requiring MV were generally younger, with a higher proportion of female patients. The onset of dengue shock occurred earlier in the MV group, with a median of 4 days from the onset of illness. These patients also exhibited a greater DSS severity than those who did not require MV. Severe bleeding complications and elevated INR levels were more prevalent in the MV group. Patients in the MV group had higher systolic and diastolic shock indices and required more vasopressor support, as indicated by a significantly higher vasoactive inotropic score. Liver injury was more pronounced in the MV group, as evidenced by the significantly elevated levels of AST, ALT, and serum lactate. Severe transaminitis was more common in the MV group. Hematological parameters also differed between the groups; patients requiring MV had lower hemoglobin levels, peak hematocrit, nadir hematocrit, and significantly lower platelet counts. Respiratory distress was more pronounced in the MV group, as indicated by a higher respiratory rate, with marked differences in PCO2, PO2, and serum bicarbonate levels compared with the non-MV group. In terms of fluid management, the MV group received a greater volume of intravenous fluids both from the referral hospital and during the first 24 h of PICU admission. They also had higher colloid-to-crystalloid fluid infusion ratios. Notably, patients with DSS who required MV had worse outcomes, including significantly higher fatality rates and longer hospital stays, than those who did not require MV.

### Performance of supervised models for mechanical ventilation in DSS patients

Table 2 summarizes the performance of various supervised machine learning models employed to estimate the risk of MV in patients with dengue shock syndrome. The LR achieved an area under the curve of 0.949, with a sensitivity of 0.995, modest specificity of 0.597, precision of 0.941, F1 score of 0.732, and accuracy of 0.942. This LR model showed high sensitivity and accuracy, making it reliable for identifying true positives. However, its markedly lower specificity indicated a higher rate of false positives. RF had an AUC of 0.959, a sensitivity of 0.690, a specificity of 0.979, a precision of 0.841, an F1 score of 0.755, and an accuracy of 0.938. The RF model showed strong performance with high AUC, specificity, and accuracy. The SVM model had an AUC of 0.963, a sensitivity of 0.860, a specificity of 0.959, a precision of 0.725, an F1 score of 0.787, and an accuracy of 0.948. The SVM exhibited excellent performance with high sensitivity, specificity, and precision, making it a robust model for prediction. The highest AUC indicates that the SVM model has a strong discrimination power. The KNN model had an AUC of 0.794, a sensitivity of 0.686, a specificity of 0.902, a precision of 0.875, an F1 score of 0.769, and an accuracy of 0.794. The XGBoost achieved an AUC of 0.824, sensitivity, specificity, and accuracy of 0.824, precision and F1 score of 0.823. The XGBoost has demonstrated potential for predicting MV in pediatric patients with DSS.

**Table 1. Baseline clinical and laboratory characteristics of study participants on PICU admission and clinical outcomes at discharge (N = 1,278).**

| Characteristics | Mechanical ventilation (n = 170) | Non-mechanical ventilation (n = 1,108) | p-value |
|---|---|---|---|
| Age (years) | 6.9 (4–9) | 8.4 (5.7–10.9) | < 0.001 |
| Female patients (%) | 95 (56) | 524 (47) | 0.045 |
| Body mass index (kg/m$^2$) | 18.4 (16–22) | 18.1 (15.5–21.4) | < 0.01 |
| Underlying diseases (%) | 13 (7.7) | 67 (6.1) | 0.323 |
| Day of occurrence of dengue shock since onset of fever (days) | 4 (4–5) | 5 (4–5) | |
| Grading of DSS severity (%) | | | < 0.001 |
| Compensated DSS | 137 (81) | 1,034 (93) | |
| Decompensated DSS | 33 (19) | 74 (7) | |
| Severe bleeding (%) | 77 (45) | 18 (1.6) | < 0.001 |
| Respiratory rate (/min) | 30 (25–35) | 25 (22–30) | |
| Systolic shock index (bpm/mmHg) | 1.45 (1.2–1.71) | 1.29 (1.11–1.44) | < 0.001 |
| Diastolic shock index (bpm/mmHg) | 2.07 (1.71–2.67) | 1.65 (1.43–2.0) | < 0.001 |
| White blood cell count (x 10$^9$/L) | 6.35 (4.2–10.3) | 4.6 (3.3–6.5) | 0.764 |
| Hemoglobin (g/dL) | 13.5 (11.8–15) | 15 (13.7–16.2) | 0.015 |
| Peak hematocrit (%) | 47 (41–50) | 48 (45–51) | < 0.001 |
| Nadir hematocrit (%) | 36 (30–41) | 38 (35–41) | < 0.001 |
| Platelet counts (x 10$^9$/L) | 30 (17.6–47) | 37 (24–56.8) | 0.014 |
| Aspartate aminotransferase (IU/L) | 588 (162–2020) | 140 (84–275) | < 0.001 |
| Alanine aminotransferase (IU/L) | 253 (56–768) | 62 (33–132) | < 0.001 |
| Severe transaminitis (%) | 68 (40) | 62 (5.6) | < 0.001 |
| International normalized ratio (INR) | 1.85 (1.44–2.49) | 1.19 (1.09–1.37) | < 0.001 |
| Serum lactate (mmol/L) | 3.2 (1.9–6.5) | 2.3 (1.6–3.0) | < 0.001 |
| Serum creatinine (mmol/L) | 52 (42–66) | 52 (45–59) | 0.001 |
| Arterial blood gas analysis | | | |
| pH | 7.39 (7.31–7.45) | 7.44 (7.41–7.48) | 0.32 |
| PCO$_2$ (mmHg) | 27.4 (21.7–33) | 26 (22.1–29.7) | 0.01 |
| PO$_2$ with oxygen support (mmHg) | 138 (93–174) | 106 (61–144) | < 0.001 |
| Bicarbonate (mEq/L) | 16 (13.8–18.6) | 17.5 (15–19.7) | 0.03 |
| Cumulative fluid infused from referral hospitals and 24h PICU admission (mL/kg) | 258 (187–357) | 127 (101–163) | < 0.001 |
| Colloid-to-crystalloid infusion ratio | 3.1 (1.1–6.9) | 0.6 (0–1.7) | < 0.001 |
| Vasopressor support during first 24h (%) | 92 (54) | 10 (0.9) | < 0.001 |
| Vasoactive inotropic score | 10 (0–30) | 0 (0–0) | < 0.001 |
| Length of hospital stay (days) | 12 (9–18) | 4 (3–5) | < 0.001 |
| Fatal outcome (%) | 38 (22.4) | 1 (0.1) | < 0.001 |

Summary statistics are presented as median (interquartile range, IQR) for continuous variables and frequency (%) for categorical variables. Two-sided, two-sample t-tests were used for the comparison of continuous variables, and the chi-square test (adjusted Fisher exact test) was used for categorical group comparisons.
Abbreviations: DSS, dengue shock syndrome; PICU, pediatric intensive care unit.

## Variables of importance from the Random Forest model

The RF model identified the most significant features contributing to the prediction of mechanical ventilation (MV), as shown in **Fig 2**. These features included the cumulative volume of fluid infused from the referral hospital and during the first 24 hours of PICU admission, a higher colloid-to-crystalloid fluid infusion ratio, low platelet counts (< 20 x 10$^9$/L), severe bleeding, vasoactive inotropic score (VIS) > 30, age < 5 years, female patients, and early onset of DSS.

**Table 2. Performance of supervised models to estimate the risk of mechanical ventilation in patients with dengue shock syndrome (N = 1,278).**

| Models | AUC | Sensitivity | Specificity | Precision | F1 score | Accuracy |
|---|---|---|---|---|---|---|
| LR | 0.949 | 0.995 | 0.597 | 0.941 | 0.732 | 0.942 |
| RF | 0.959 | 0.690 | 0.979 | 0.841 | 0.755 | 0.938 |
| SVM | 0.963 | 0.860 | 0.959 | 0.725 | 0.787 | 0.948 |
| KNN | 0.794 | 0.686 | 0.902 | 0.875 | 0.769 | 0.794 |
| XGBoost | 0.824 | 0.824 | 0.824 | 0.823 | 0.823 | 0.824 |

AUC, Area under curve; LR, Logistic Regression; RF, Random Forest; KNN, K-Nearest Neighbors; SVM, Support Vector Machine; XGBoost, eXtreme Gradient Boosting.

### The Shapley Additive Explanations (SHAP) model

The SHAP plot highlights the most important variables contributing to the risk of mechanical ventilation in hospitalized children with dengue shock syndrome (**Fig 3**). Most importantly, all predictors from the SVM model demonstrated the utmost significance in the SHAP model. Notably, the cumulative amount of infused fluid had the highest SHAP value, indicating that it was the most influential predictor of mechanical ventilation. A higher colloidal-to-crystalloid fluid infusion ratio was associated with an increased need for mechanical ventilation in patients with DSS. The vasoactive inotropic score reflects the intensity of vasoactive medications used to support hemodynamics and cardiac function. DSS patients with a high VIS (> 30) exhibited a markedly increased risk of requiring MV, indicating critical cardiovascular dysfunction. Severe bleeding, a serious complication of dengue, was also a significant predictor

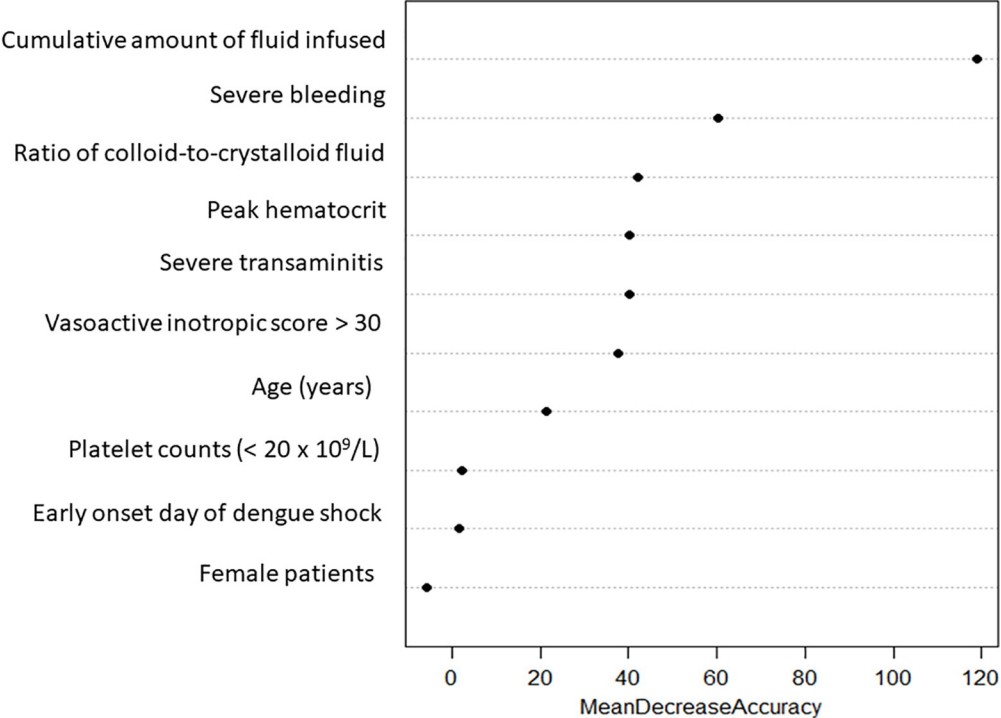

**Fig 2. The ten most significant features of the Random Forest model based on classification accuracy.**

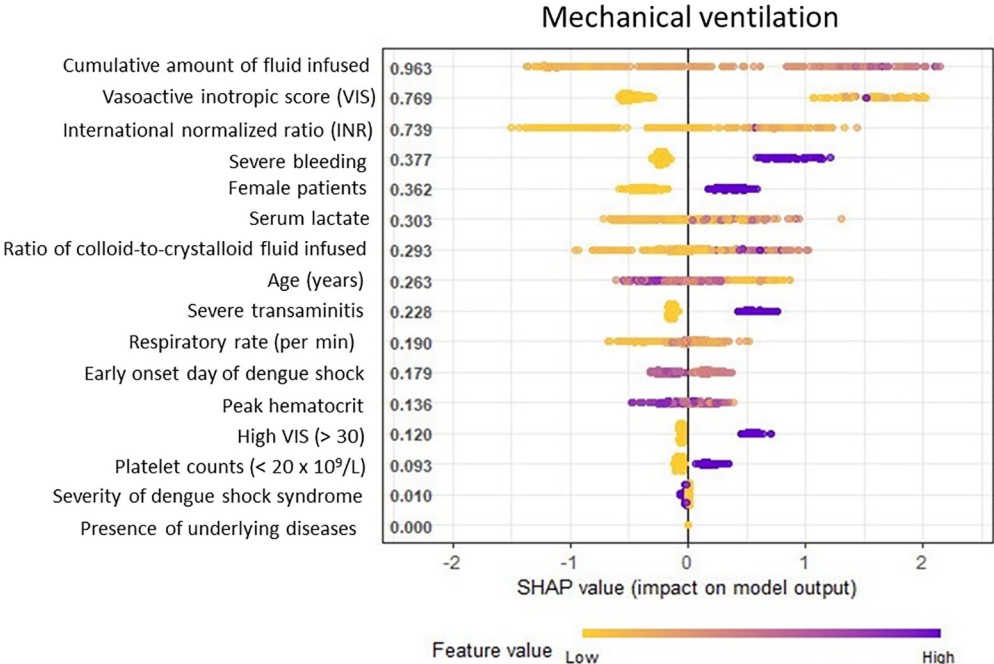

**Fig 3. The SHAP interpretation of the clinically important variables in patients with dengue shock syndrome on mechanical ventilation.** All ten features identified by the SVM model showed high significance in the SHAP model. However, the remaining variables had less impact on mechanical ventilation.

of MV. Other important contributors to the SHAP model included severe transaminitis, low platelet counts ($< 20 \times 10^9$/L), and elevated INR, which were strongly associated with the need for mechanical ventilation. Increased serum lactate levels, indicative of severe metabolic acidosis and liver dysfunction, also predicted the requirement for MV. In addition, age was another significant factor, with younger children showing a higher risk of MV. Female patients were more likely to require MV support than male patients, making sex an important predictive factor. An increased respiratory rate upon admission, indicating respiratory distress, was predictive of MV. Additionally, early onset of DSS was identified as a risk factor for mechanical ventilation. Elevated hematocrit levels were found to be an important predictor in the SHAP model and were demonstrated to be a protective factor against mechanical ventilation.

## Final predictive model, internal validation, and calibration

The best-performing model for predicting MV, incorporating the ten most significant features from the SVM and SHAP models, was internally validated using complete-case analysis and the bootstrap method. As shown in **Fig 4**, the final predictive model demonstrated strong consistency between the predicted values and actual observations, indicating excellent calibration of both the slope and intercept, as well as a low Brier score (0.04). These results confirm that the final predictive model is robust in predicting severe respiratory failure requiring MV in pediatric patients with dengue shock syndrome.

## Risk score for mechanical ventilation among patients with DSS

A complete-case analysis multivariable logistic model (n = 1,268 participants) utilizing the ten most significant features identified by the supervised models was used to construct a risk-

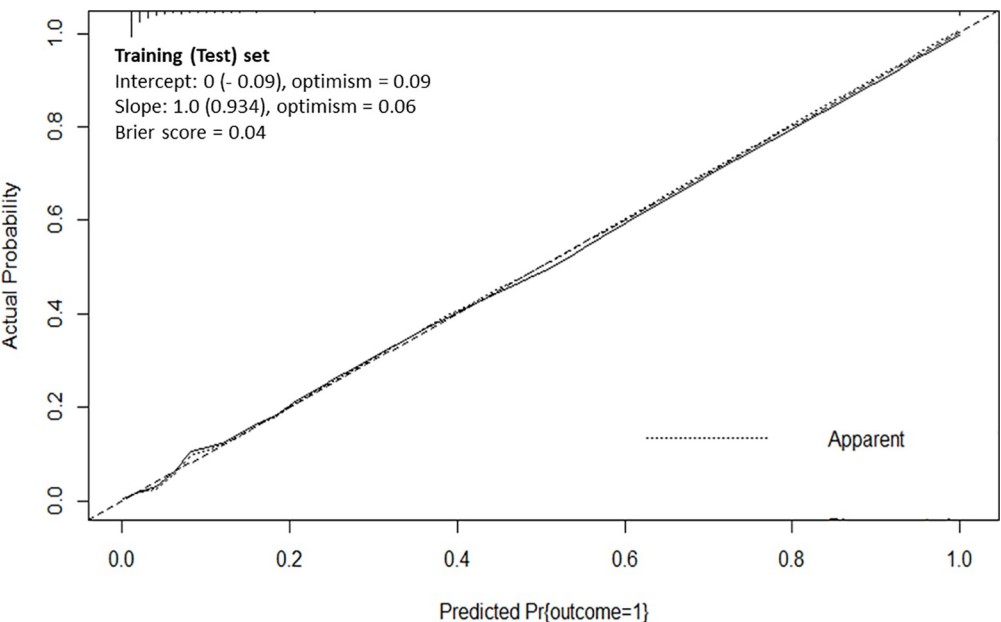

**Fig 4. Calibration plot showing high consistency between the predicted values (x-axis) and actual observation data (y-axis).**

scoring system (**Table 3**). The risk score was developed based on the subcoefficients of these ten predictors. The total points ranged from -2 to 46, corresponding to a probability of mechanical ventilation ranging from 0% to 100%. As shown in **Fig 5,** the risk of mechanical ventilation in hospitalized children with DSS increased proportionately with an elevated risk score.

## Discussion

Patients with severe dengue who experience respiratory failure and require mechanical ventilation have a significantly higher mortality rate [8–10]. High proportions of severe dengue patients with critical respiratory distress requiring mechanical ventilation support have been reported, ranging from approximately 25% to 35% in pediatric and adult cohorts, respectively [4,10]. In this study, 13.3% of the children with DSS required MV support, and a dramatically higher fatality rate was observed in the MV group than in the non-MV group (22.4% vs. 0.1%). Recognizing the prognostic indicators associated with MV in patients with DSS is crucial for optimizing the management protocols and improving survival outcomes.

Although several studies have identified respiratory failure as a critical predictor of fatality in severe dengue [7,8,15,16], data comprehensively examining the determinants of MV support in DSS patients remain limited [3,8,18]. Previous research has identified several risk factors for MV in DSS, including female sex, obesity, large volume of intravenous fluid infusion, severe transaminitis and bleeding [3,8,18]. However, previous studies were constrained by small sample sizes or lacked robust statistical methods to determine the definitive prognosticators of MV [3,8,18–20]. Given the increasing popularity of machine learning owing to its robust modeling capabilities in various diseases [12–16], we employed supervised models to identify the key prognosticators of MV in children with DSS. Using predefined predictive covariates obtained from clinical knowledge and medical literature, we applied the LASSO method to prevent overfitting. The ten most significant features identified showed strong

**Table 3. Risk score to predict mechanical ventilation among children with dengue shock syndrome during the first 24 hours of PICU admission.**

| Predictors | Subcategory | Reference values | Subcoefficients | SE | Risk score |
|---|---|---|---|---|---|
| Age (years) | ≥ 5 | Reference | 0 | - | 0 |
| | < 5 | - | 1.143 | 0.321 | 4 |
| Female patients | No | Reference | 0 | - | 0 |
| | Yes | - | 0.807 | 0.290 | 3 |
| Cumulative amount of fluid infused (mL/kg) | ≤ 180 | Reference | 0 | - | 0 |
| | > 180 | - | 2.618 | 0.320 | 10 |
| Colloid-to-crystalloid fluid infusion ratio | < 1.6 | Reference | 0 | - | 0 |
| | ≥ 1.6 | - | 0.873 | 0.297 | 3 |
| Platelet counts ($< 20 \times 10^9$/L) | No | Reference | 0 | - | 0 |
| | Yes | - | 0.264 | 0.329 | 1 |
| Severe bleeding | No | Reference | 0 | - | 0 |
| | Yes | - | 3.171 | 0.414 | 12 |
| + 01-log2-increase in VIS during first 24h of admission | No | Reference | 0 | - | 0 |
| | Yes | - | 1.138 | 0.148 | 4 |
| Severe transaminitis | No | Reference | 0 | - | 0 |
| | Yes | - | 1.789 | 0.385 | 7 |
| Onset day of shock | > day 4 | Reference | 0 | - | 0 |
| | ≤ day 4 | - | 0.510 | 0.290 | 2 |
| Peak hematocrit (%) | ≤ 48% | Reference | 0 | - | 0 |
| | > 48% | - | - 0.447 | 0.287 | -2 |
| *Range of risk score* | From -2 to 46 | | | | |

Sub-coefficients derived from the slopes of the multivariable logistic model (complete-case analysis); Abbreviations: PICU, pediatric intensive care unit; SE, standard errors of coefficients; VIS, vasoactive inotropic score.

performance in the supervised models and SHAP analysis, aligning with previous reports [4,8,18–20].

The predictive factors identified in this study, including a large cumulative fluid infusion from the referral hospital and during the first 24 h of PICU admission, a high colloid-to-crystalloid infusion ratio, severe bleeding, and low platelet counts ($< 20 \times 10^9$/L), particularly in patients requiring platelet transfusion, severe transaminitis, and an increased vasoactive inotropic score, were strong predictors of MV in patients with DSS. Additionally, younger age and female sex are associated with severe dengue, as reported in systematic reviews and meta-analyses [21,22]. Most importantly, dengue-related complications, including acute respiratory disease, severe transaminitis and bleeding, and a high VIS ($> 30$) have been shown to have high prognostic values for dengue-associated fatality [4,19,20]. Therefore, these data provide the basis for our predetermined covariates and the use of mechanical ventilation as a surrogate outcome for dengue-related mortality in patients with dengue shock syndrome. Notably, other covariates, including obesity, underlying diseases, and serum lactate levels, which have been reported as important risk factors for adverse clinical outcomes in patients with DSS, were not significant in LASSO and machine learning analyses [4,22].

Most significantly, excessive fluid resuscitation has a negative impact on clinical outcomes in PICU-admitted patients as well as in patients with severe dengue [5,23–25]. Each 1% excess in fluid resuscitation increases the risk of mortality by 6%, primarily by prolonging mechanical ventilation and inducing acute kidney injury [23]. Excess fluid resuscitation results in increased pressure in multiple body compartments, particularly the abdominal and thoracic cavities. This can lead to multiorgan failure due to prolonged, hypovolemic, and obstructive

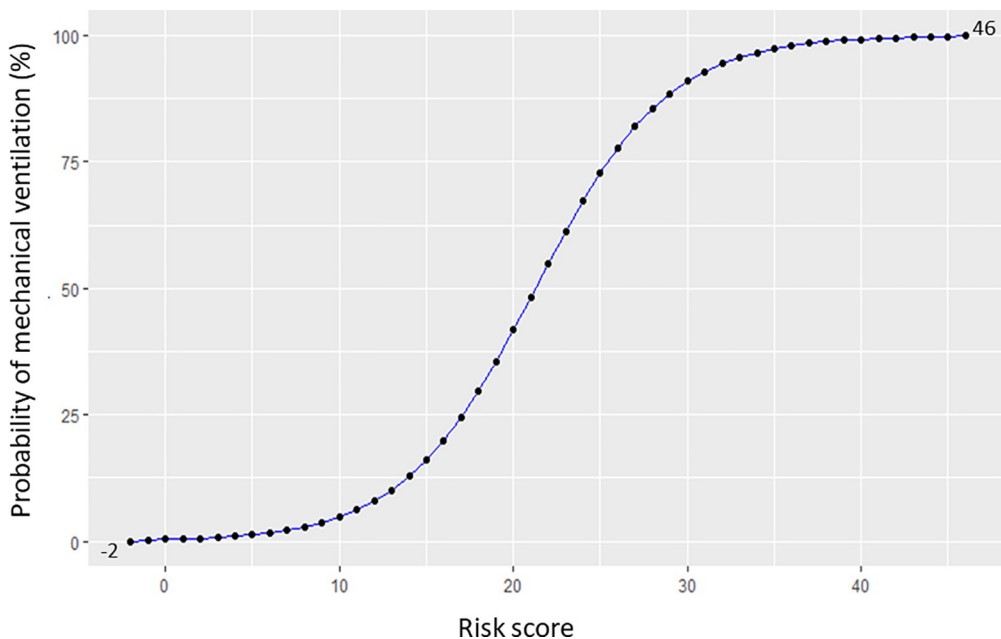

**Fig 5. A simplified risk score chart for estimating the risk of mechanical ventilation in children with dengue shock syndrome.** The total points ranged from -2 to 46, corresponding to a probability of mechanical ventilation ranging from 0% to 100%.

dengue shock when patients with severe DSS are resuscitated with a large volume of intravenous fluid [9,11]. Owing to the multiple adverse effects, macromolecular solutions are restrictively used in shock resuscitation, except for patients with DSS [26,27]. The use of colloidal solutions is crucial for patients with DSS when the initial resuscitation with crystalloid fluid shows a poor response. The judicious use of colloids is recommended, with a minimal amount of fluid infused to maintain hemodynamic stability. Notably, low serum albumin levels have been shown to be associated with progression to severe dengue; therefore, albumin solution should be administered to patients with severe DSS accompanied by unstable hemodynamic conditions [2,28,29]. In this study, an elevated colloid-to-crystalloid fluid infusion ratio (> 1.6) significantly predicted the need for mechanical ventilation. In clinical practice, a cut-off ratio of > 1.6 is a good indicator of the appropriate time point for switching from colloid solutions to more beneficial fluids such as albumin and fresh frozen plasma. Markedly, severe bleeding is a well-recognized predictor of adverse outcomes in DSS patients [4,30]. Severe transaminitis and coagulation dysfunction are frequently observed in patients with DSS. Therefore, colloid infusion in patients with severe DSS can potentially aggravate coagulation disorders and bleeding, thereby increasing the demand for blood product transfusions [27]. Furthermore, iatrogenic hemodilution in patients receiving large volumes of colloid solutions may result in a paradoxical decrease in oxygen delivery owing to a decrease in hemoglobin concentration, which can increase unnecessary transfusion of blood products [31]. Another point to be highlighted is that elevated hematocrit is associated with hemoconcentration in patients with DSS, and fluid supplementation is necessary in this regard [2]. Notably, the most important goal of treating DSS is to infuse a minimum amount of intravenous fluid to ensure adequate tissue perfusion and urine output of > 0.5 ml/kg/h [32]. When patients with elevated hematocrit levels present with good hemoperfusion, the excessive administration of fluid targeted at reducing and/or maintaining low hematocrit levels may lead to poor clinical

outcomes. Importantly, increased hematocrit levels have been shown to be a protective factor in reducing the risk of MV [3]. Therefore, emphasis is placed on sustaining a minimal amount of fluid to achieve good hemoperfusion of vital organs and maintain urine output > 0.5 ml/kg/h. In the presence of signs of adequate hemoperfusion, reducing the rate of intravenous fluid infusion is recommended [32].

The final prediction model was developed based on the ten most important features identified by the LASSO and machine learning models. On this basis, we constructed a robust risk-scoring system that is helpful for clinicians in the PICU for the early recognition of DSS patients requiring mechanical ventilation support. The total risk score ranged from minus 2 to 46 points. Notably, an elevated hematocrit > 48% is considered a protective factor, corresponding to a deduction of 2 points in the risk-scoring system. As shown in **Fig 5**, the risk score chart showed that DSS patients with a risk score of 14–15 had roughly 10% progression to respiratory failure requiring MV support. A higher risk score indicates an increased probability of MV support. Hence, DSS patients with a risk score of 21 had a 50% probability of receiving MV support, and those with a risk score of 38–39 had a 99% probability of receiving MV. Given the proportional relationship between MV requirement and mortality risk in hospitalized children with DSS, this risk-scoring system is a valuable tool for improving clinical management. Further external and prospective validation in other cohorts will enhance the generalizability of these findings.

Dengue shock syndrome has been reported to have fatality rates ranging from 5.3% to 25.6% in all-type DSS-experiencing children admitted to the PICU [4,5]. Similarly, DSS patients who develop critical respiratory failure account for markedly elevated mortality rates in pediatric and adult cohort studies [6,8,9]. The combination of these dengue-associated complications has predominantly contributed to significantly increased fatalities among hospitalized dengue patients, as noted in our recent cohort of severely prolonged DSS patients undergoing mechanical ventilation (death rate of approximately 36%) [9]. Mechanical ventilation is a frequently observed complication among critically ill patients in the late critical stage of severe dengue infection [11]. Therefore, this study has great clinical significance. Rapid identification of patients with high-risk DSS for MV support at the time of PICU admission could help prioritize interventions that can significantly enhance clinical and survival outcomes.

This study has several strengths. The predictive model for mechanical ventilation was based on a relatively large sample of hospitalized children with DSS. The most significant features were well-defined using profound clinical knowledge and robust machine learning methods. Internal validation of the final predictive model demonstrated high discrimination and calibration. However, this study had several limitations inherent to its single-center retrospective cohort design and unstandardized collection of clinical and laboratory data during hospital admission. The most significant bias was the missing information on serum lactate levels, previously reported as a prognostic indicator of death. Nevertheless, this variable was appropriately managed using the multiple imputation method.

## Conclusion

We developed robust machine learning-based models to estimate the risk of mechanical ventilation in hospitalized children with dengue shock syndrome. The internal validation of the predictive model demonstrated high performance. Based on the most significant features identified at the time of admission, we constructed a simple risk-scoring system. This scoring system is a valuable tool for clinicians, aiding in the bedside management of DSS patients.

## Supporting information

**S1 Checklist. TRIPOD checklist.**
(DOCX)

**S1 File. Indications for mechanical ventilation support in children with severe DSS.**
(DOCX)

**S1 Table. Data variables used in the machine learning modelling.**
(DOCX)

**S1 Fig. The Pareto chart of missing values of the initial set of 28 candidate variables.**
(TIF)

## Acknowledgments

We are grateful to the patients and the administrative staffs for their participation in this study.

## Author Contributions

**Conceptualization:** Nguyen Tat Thanh, Vo Thanh Luan, Vu Thien.

**Data curation:** Nguyen Tat Thanh, Vo Thanh Luan.

**Formal analysis:** Nguyen Tat Thanh, Vu Thien.

**Funding acquisition:** Vo Thanh Luan, Do Chau Viet.

**Investigation:** Nguyen Tat Thanh, Vo Thanh Luan, Do Chau Viet.

**Methodology:** Nguyen Tat Thanh, Vo Thanh Luan, Vu Thien.

**Project administration:** Vo Thanh Luan, Do Chau Viet, Trinh Huu Tung.

**Resources:** Vo Thanh Luan, Trinh Huu Tung.

**Supervision:** Vo Thanh Luan, Do Chau Viet, Trinh Huu Tung.

**Writing – original draft:** Nguyen Tat Thanh, Vo Thanh Luan, Vu Thien.

**Writing – review & editing:** Nguyen Tat Thanh, Vo Thanh Luan, Do Chau Viet, Trinh Huu Tung, Vu Thien.

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
