## [Decision Letter · Decision Letter 0]

9 Sep 2024

PONE-D-24-33301A machine learning-based risk score for prediction of mechanical ventilation in children with dengue shock syndrome: A retrospective cohort studyPLOS ONE

Dear Dr. Nguyen,

Thank you for submitting your manuscript to PLOS ONE. After careful consideration, we feel that it has merit but does not fully meet PLOS ONE’s publication criteria as it currently stands. Therefore, we invite you to submit a revised version of the manuscript that addresses the points raised during the review process.

**ACADEMIC EDITOR: ** 1. Please ensure that extensive language editing is performed prior to submission of the revised manuscript.**2.** Authors use medical terms that are not correct in many places eg: severe hepatic transaminases - this should be severe transaminitis3. It is not clear whether this risk score is to be used for patients with respiratory failure or all DSS. clarify.4. Inclusion criteria is not clear.

- Was the model performed on all DSS, all DHF or DHF/DSS with respiratory failure?

- Were they laboratory-confirmed dengue cases?

5. What were the exclusion criteria? Did you exclude co-infection causing pneumonia and respiratory failure?

7. Conclusion of the abstract – vague. To manage patients with respiratory failure and predict a poor outcome? Is it actually to predict MV in all severe dengue? 

We look forward to receiving your revised manuscript.

Kind regards,

Nilanka Perera, MD, PhD

Academic Editor

PLOS ONE

Reviewers' comments:

Reviewer's Responses to Questions

**Comments to the Author**

1. Is the manuscript technically sound, and do the data support the conclusions?

Reviewer #1: Yes

Reviewer #2: Yes

2. Has the statistical analysis been performed appropriately and rigorously? 

Reviewer #1: No

Reviewer #2: I Don't Know

3. Have the authors made all data underlying the findings in their manuscript fully available?

Reviewer #1: Yes

Reviewer #2: Yes

4. Is the manuscript presented in an intelligible fashion and written in standard English?

Reviewer #1: Yes

Reviewer #2: No

5. Review Comments to the Author

Reviewer #1: I have the following feedback on the write-up. The authors have applied several machine learning models to a real world dataset. Out of many variable, a selected set of variables were chosen for the analysis.

-There are some grammatical errors and typos. Read the manuscript thoroughly and correct them.

-In the abstract, it says SHAP model explained the significant clinical predictors. However, authors need to mention the machine learning model they used with SHAP model, eg: RF or SVM etc.

-Authors used variable importance plot with RF model and SVM or RF (not clear) with SHAP to identify the importance features. Authors need to mention why they considered two methods, or why they preffered SHAP method over variable importance given by RF.

-In line 292, Final predictive model, internal validation and calibration section, it's better to include the name of the final model (RF or SVM etc).

-In line 233, authors mentioned that Logistic regression (LR) model has higher rate of false positives. Then they developed a risk score for mechanical ventilation among patients with DSS using LR model in line 304 under "Risk score for mechanical ventilation among patients with DSS" section, which is not justifiable.

In this study, there's a significant class imbalance between MV (170) and non-MV groups (1108). Authors can try using class balancing techniques to see if they can further improve the precision and recall along with the accuracies. In addition to that, LASSO models work best for variable selection when there's multicollinearity and the usual practice is to check for multicollinearity before applying LASSO models.Out of many variable, a selected set of variables were chosen for the analysis using some ad-hoc method and it is recommended to use a proper variable selection method (from all non-missing variables)

Reviewer #2: This is an interesting analysis that attempts to incorporate artificial intelligence into clinical practice. However, the flow of information presented in the text can be improved.

I would like to suggest the following.

Introduction section

1. The authors should verify the objective of this study. In lines 106 and 127, different aims have been mentioned.

Methods section

2. Contradictory information on the study population. Please verify ( refer to line 129).

3. It is unclear how participants were recruited from the database and who was eligible for this particular study. Please explain.

4. Out of 82 variables in eligible participants, 28 variables were initially selected for analysis. Describe the rationale for selecting only some variables for analysis (lines 139 & 140).

5. Was there a uniformity of criteria used for starting mechanical ventilation at the PICU?

The authors should clarify this. If different clinicians used different criteria, there is a potential for bias in this analysis.

6. Data on participant numbers should also be included in the methods section (refer lines 203 & 204).

Results section

1. The authors have given general statements in the results section. For example “ patients who required MV were generally younger and more females than males…”. Please describe in terms of mean or median age and sex percentage (refer line 206).

2. Please define the terms used in the text – critical bleeding, severe transaminitis, etc.

3. How did the authors grade the severity of dengue infection?

4. Table 1

• The naming of the 2nd and 3rd columns is confusing. It sounds as if one group received a form of non-mechanical ventilation. Therefore, name them appropriately. For example, MV required and MV not required.

• Mention the statistical tests used to derive p value in table 1.

5. Please describe how sub-coefficients were used to derive a risk score (refer line 307).

6. Explain how to use risk score practically.

7. Table 3

• Define severe bleeding and severe transaminitis in the main text as well. ( There are defined in supporting documents)

• It is necessary to decide on a cut-off value for severe transaminitis to derive the risk score.

Discussion section

1. In Table 3, it is mentioned thrombocytopenia is a risk factor but, in the text, it is said thrombocytopenia needing platelet transfusions is a risk factor. Verify if thrombocytopenia per se or thrombocytopenia requiring platelet transfusions was a risk factor for MV.

2. Have the platelet transfusions caused respiratory failure by fluid overload?

3. Investigating if blood transfusions were a risk factor for MV may be interesting.

4. Previously identified risk factors (such as obesity and lactate levels) for adverse outcomes for DSS were not significant in your analysis. Do you have any explanation for this difference? (lines 359 to 362).

5. Line 374 & 375 “Albumin solution should be administered to patients with severe DSS accompanied with unstable hemodynamic conditions” – Is this evidence-based?

6. Do you have any explanation for why increased hematocrit is a protective factor for MV?

7. Validation of this score with prospective studies is necessary before recommending this score in clinical practice (line 408).

General other comments

1. Different units for platelet counts have been used. Please stick to one unit.

2. Some spelling and grammar mistakes were noted, and language editing is recommended.

6. PLOS authors have the option to publish the peer review history of their article (what does this mean?). If published, this will include your full peer review and any attached files.

Reviewer #1: No

Reviewer #2: No

---

## [Author Response · Author response to Decision Letter 0]

19 Sep 2024

Response letter to reviewers

Dear Editors-in-Chief of the PLOS ONE,

We are very thankful for your insightful comments for our submitted manuscript. 

MANUSCRIPT # PONE-D-24-33301, entitled "A machine learning-based risk score for prediction of mechanical ventilation in children with dengue shock syndrome: A retrospective cohort study" 

We are thankful for all peer-review points. These are our responses to all the reviewers’ comments.

ACADEMIC EDITOR:

1. Please ensure that extensive language editing is performed prior to submission of the revised manuscript.

Response: Yes, we are thankful for your comments. We have checked the language editing in the revision manuscript and resubmitted the latest version as appropriate.

2. Authors use medical terms that are not correct in many places 

eg: severe hepatic transaminases - this should be severe transaminitis

Response: We are thankful for your comments, and appropriate amendments have been made to the revised manuscript from your suggestions. 

3. It is not clear whether this risk score is to be used for patients with respiratory failure or all DSS. clarify.

Response: We would like to clarify that this risk score of mechanical ventilation (main study outcome) will be used for all patients with dengue shock syndrome (DSS) who were admitted to the PICU. Among the DSS patients, a certain proportion of children with DSS (170/1278, 13.3%) developed severe respiratory failure that required MV support. 

This information has been included in the revision manuscript.

4. Inclusion criteria is not clear.

- Was the model performed on all DSS, all DHF or DHF/DSS with respiratory failure?

- Were they laboratory-confirmed dengue cases?

Response: We thank for your comment, and we have made appropriate amendments as follows:

Lines 128-129 in the revision manuscript.

“Eligibility criteria were defined as age below 18 years, laboratory-confirmed dengue infection, and the presence of dengue shock syndrome.” 

It is notable the denominator (population) are all PICU-admitted children had laboratory-confirmed dengue infection and DSS, with / without respiratory distress on PICU admission. 

5. What were the exclusion criteria? Did you exclude co-infection causing pneumonia and respiratory failure?

Response: We are thankful for your comments and have made clarification and amendment as appropriate. 

We have made amendments to clarify this point as follows: 

“The exclusion criteria were a lack of serologically confirmed dengue infection and missing data for variables of interest ( ≥ 50%).”

Lines 129-130, in the revision manuscript.

In this study, we aimed to increase the study representativeness and generalizability of the prognostic model. When we developed a baseline prognostic model for mechanical ventilation during 24h of PICU admission, we did not exclude co-infection-causing pneumonia related to mechanical ventilation. Because the ventilator-associated pneumonia (VAP) commonly occurs ≥ 48h -72h after PICU admission (relatively far beyond the baseline timepoint-24h PICU admission)

Additionally, we did not exclude patients with DSS who experienced respiratory failure when they were admitted to the PICU. This is because all referred DSS patients were commonly treated previously within 12h-24h from the province hospital and then transferred to our hospital. Some patients might have respiratory failure and others did not. The most common cause of respiratory failure in the referred DSS patients was due to fluid overload and other dengue-related complications, such as severe bleeding, acute liver failure. 

Therefore, with these approaches, we could increase the representativeness and generalizability. 

7. Conclusion of the abstract – vague. To manage patients with respiratory failure and predict a poor outcome? Is it actually to predict MV in all severe dengue? 

Response: We are thankful for your comments. We have made appropriate amendments for clarification, as follows:

“We developed a robust machine learning-based risk score to estimate the need for MV in hospitalized children with DSS.”

Lines 60-61, in the revision manuscript.

Reviewer #1:

I have the following feedback on the write-up. The authors have applied several machine learning models to a real world dataset. Out of many variable, a selected set of variables were chosen for the analysis.

-There are some grammatical errors and typos. Read the manuscript thoroughly and correct them.

Response: We are thankful for your comments, and we will make appropriate amendments.

-In the abstract, it says SHAP model explained the significant clinical predictors. However, authors need to mention the machine learning model they used with SHAP model, eg: RF or SVM etc.

Response: Thank for your comment. In this study, we utilized several supervised machine

learning models, including Logistic Regression (LR), Random Forest (RF), Support Vector

Machine (SVM), k-Nearest Neighbors (KNN), and Extreme Gradient Boosting (XGBoost), to

predict the need for mechanical ventilation in pediatric patients with dengue shock syndrome. Upon evaluating these models, we found that SVM and RF models demonstrated very good overall performance (AUC, Sensitivity, Specificity, Precision, F1 score, and Accuracy). 

We based the optimal performance metrics on determining the SVM model to be additionally analyzed with SHAP. 

We add this information in the revised manuscript, Lines 298-300

- Authors used variable importance plot with RF model and SVM or RF (not clear) with SHAP to identify the importance features. Authors need to mention why they considered two methods, or why they preferred SHAP method over variable importance given by RF.

Response: Currently, there are various supervised models, and it is well noted that under different sampling conditions that one supervised model may be better than other models and vice versa. 

- Balancing all performance metrics, RF and SVM showed the two best performance with AUC of 0.96 and high levels of precision and accuracy.

- In addition, the SVM demonstrated slightly better performance than the RF model, in terms of higher sensitivity.

Therefore, we finally determined the SVM as the best performing model. However, the RF model is also considered competitive. 

Additionally, SHAP analysis was employed to provide a more detailed and interpretable explanation of how each feature influenced individual predictions. SHAP was preferred because it allows for a consistent and individualized explanation of feature importance across multiple models, offering more granular insights. 

We applied the SHAP methodology based on the best supervised model (SVM) for studying the explanation of the fully predefined covariables (also including the 10 most important features identified). In the SHAP model, the 10 features from the SVM model outperformed the remaining risk factors for MV. 

Additionally, these are examples to illustrate the point that SHAP values can provide global insights into the feature importance and local explanations of individual predictions of each covariate. 

SHAP for Age: 

As can be seen from this chart, patients with DSS aged < 5 years have an increased risk of MV. The SHAP model can provide us with the distribution of age variable and an explanation for age in the overall prediction of outcome (MV). 

Further example for SHAP model with hematocrit (HCT) variable in this study:

It can be seen from this SHAP plot that HCT > 48% corresponds to low SHAP values, indicating having a lower risk of MV.

=> Therefore, with all these supported points, the SHAP model was preferred over other supervised models (SVM, RF, KNN,….).

-In line 292, Final predictive model, internal validation and calibration section, it's better to include the name of the final model (RF or SVM etc).

Response: We are thankful for your comments. 

We used the ten most important predefined covariates in the final SVM model based on its highest performance, which was further analyzed with internal validation and calibration.

We have clarified these points as you suggested, in the revision. 

Lines 273-274, in the revision manuscript

“All the predictors from the SVM model demonstrated the utmost significance in the SHAP model”

Lines 298-300, in the revision manuscript

“The best performance model for MV, with the ten most significant features from the SVM and SHAP models, was internally validated using complete-case analysis and the bootstrap method.”

-In line 233, authors mentioned that Logistic regression (LR) model has higher rate of false positives. Then they developed a risk score for mechanical ventilation among patients with DSS using LR model in line 304 under "Risk score for mechanical ventilation among patients with DSS" section, which is not justifiable.

Line 233: “LR achieved an area under the curve (AUC) of 0.949, with a sensitivity of 0.995, modest specificity of 0.597, precision of 0.941, F1 score of 0.732, and accuracy of 0.942.”

Line 304: “These results confirm that the final predictive model performs well in predicting severe respiratory failure requiring MV among pediatric patients with dengue shock syndrome.”

Response: We thank for your comments. We will answer from point to point.

- We acknowledge that the logistic regression (LR) model has modest specificity (or a higher rate of false positives). This was mainly because of non-linearity of data (skewed data) in this study. Most of the continuous covariables were not normally distributed. We also tried several conventional methods for data normalization, including categorization and log-transformation. This is a well-known limitation of the conventional logistic analysis.

- However, machine learning supervised models can overcome this weakness from conventional analysis when manipulating with non-linearity data. With the similar 10 preselected covariates (used in the LR model above), the supervised machine learning models showed very good performance. In particular, the Support Vector Machine (SVM) is well-known for its capacity to analyze the non-linearity data. Furthermore, we additionally applied the SHAP methodology based on the best supervised model (SVM) for studying the explanation of the fully predefined covariables. 

- This is an example illustrating analyzing the peak hematocrit (HCT) with SHAP model: 

- It can be seen from this SHAP plot that the HCT is markedly non-linear, clearly observed HCT < 35% and HCT > 54%. Similarly, non-linearity was also significant, regarding covariables of age, INR, and vasoactive inotropic score (VIS). 

- Considering the predictive model performance evaluation, we used the criteria developed by Steyerberg EW et al. [1,2]. Key criteria for a well-performance model include (1) discrimination (C-statistic or AUC, discrimination slope), (2) Calibration in-the-large, calibration slope, (3) overall performance (Brier score), (4) reclassification and (5) clinical usefulness. Therefore, the specificity parameter is only one contributing indicator for assessing a predictive model, and it is not the main determinant for the evaluation of the model performance [1-5].

- In this study, despite modest specificity of LR model, the final C-statistic of LR model was still high (AUC = 94.9%), as well as precision and accuracy (of approximately 94%). Most importantly, the final predictive model (based on the 10 predefined covariables) for MV showed very good discrimination, calibration of slope, calibration plot, low Brier score as well as the clinical usefulness of the developed model [1,2]. In this study, a high consistency between predicted and observed data was prominently observed (as shown in the Figure 4). 

- We constructed a risk score based on logistic regression with complete-case analysis, considering its simplicity and interpretability in clinical practice. Logistic regression provides a straightforward method for computing risk scores that can be easily applied in clinical settings. Notably, we developed the risk score based on references from the Framingham risk score development, and clinical prediction model development and validation methods by Steyerberg EW and Harrell FE Jr [1-11].

References

1. Steyerberg EW, Vickers AJ, Cook NR, Gerds T, Gonen M, Obuchowski N, Pencina MJ, Kattan MW. Assessing the performance of prediction models: a framework for traditional and novel measures. Epidemiology. 2010 Jan;21(1):128-38. doi: 10.1097/EDE.0b013e3181c30fb2. PMID: 20010215; PMCID: PMC3575184.

2. Steyerberg EW, Vergouwe Y. Towards better clinical prediction models: seven steps for development and an ABCD for validation. Eur Heart J. 2014 Aug 1;35(29):1925-31. doi: 10.1093/eurheartj/ehu207. Epub 2014 Jun 4. PMID: 24898551; PMCID: PMC4155437.

3. Steyerberg EW, Eijkemans MJ, Harrell FE, Jr, Habbema JD. Prognostic modeling with logistic regression analysis: in search of a sensible strategy in small data sets. Med Decis Making. 2001;21:45–56.

4. Steyerberg EW, Harrell FE, Jr, Borsboom GJ, Eijkemans MJ, Vergouwe Y, Habbema JD. Internal validation of predictive models: efficiency of some procedures for logistic regression analysis. J Clin Epidemiol. 2001;54:774–781.

5. Steyerberg EW, Borsboom GJ, van Houwelingen HC, Eijkemans MJ, Habbema JD. Validation and updating of predictive logistic regression models: a study on sample size and shrinkage. Stat Med. 2004;23:2567–2586.

6. Harrell FE Jr, Lee KL, Califf RM, Pryor DB, Rosati RA. Regression modelling strategies for improved prognostic prediction. Stat Med. 1984 Apr-Jun;3(2):143-52. doi: 10.1002/sim.4780030207. PMID: 6463451.

7. Harrell FE. Regression Modeling Strategies: With Applications to Linear Models, Logistic Regression, and Survival Analysis. New York: Springer; 2001. 

8. Harrell FE Jr. Regression Modelling Strategies. Available from: https://www.r-project.org/nosvn/conferences/useR-2007/tutorials/harrell.html

9. Steyerberg EW, Eijkemans MJ, Harrell FE, Jr, Habbema JD. Prognostic modelling with logistic regression analysis: a comparison of selection and estimation methods in small data sets. Stat Med. 2000;19:1059–1079. 

10. Lloyd-Jones DM, Wilson PW, Larson MG, Beiser A, Leip EP, D'Agostino RB, Levy D. Framingham risk score and prediction of lifetime risk for coronary heart disease. Am J Cardiol. 2004 Jul 1;94(1):20-4. doi: 10.1016/j.amjcard.2004.03.023. PMID: 15219502

11. Berry JD, Lloyd-Jones DM, Garside DB, Greenland P. Framingham risk score and prediction of coronary heart disease death in young men. Am Heart J. 2007 Jul;154(1):80-6. doi: 10.1016/j.ahj.2007.03.042. PMID: 17584558; PMCID: PMC2279177.

In this study, there's a significant class imbalance between MV (170) and non-MV groups (1108). Authors can try using class balancing techniques to see if they can further improve the precision and recall along with the accuracies. In addition to that, LASSO models work best for variable selection when there's multicollinearity and the usual practice is to check for multicollinearity before applying LASSO models. Out of many variable, a selected set of variables were chosen for the analysis using some ad-hoc method and it is recommended to use a proper variable selection method (from all non-missing variables).

Response: 

The proportion of MV among the DSS population was 170 / 1278 (13.3%), which is considered to have relatively sufficient statistical power for predictive model development. Importantly, this figure (MV prevalence of 13.3% among DSS) truly reflects the real-life data of patients with dengue shock syndrome, that is consistent with the previously published data [1]. Therefore, we do not think that the MV imbalance is considerable in this study. We conservatively believe that it is not necessary to perform oversampling methods (SMOTE...etc). 

Also, we did not find any multicollinearity when analyzing the LASSO. Additionally, all covariates were independent. We recognized that LASSO seemed to be too stringent in variable selection, and eliminated some significant variables that are well known in disease pathogenesis and clinical importance. As stated in the manuscript, we combined and balanced different methods of LASSO, clinical knowledge, and the medical literature for preselecting the covariates. 

Reference 

[1] 

---

## [Decision Letter · Decision Letter 1]

25 Nov 2024

A machine learning-based risk score for prediction of mechanical ventilation in children with dengue shock syndrome: A retrospective cohort study

PONE-D-24-33301R1

Dear Dr. Nguyen,

We’re pleased to inform you that your manuscript has been judged scientifically suitable for publication and will be formally accepted for publication once it meets all outstanding technical requirements.

Kind regards,

Nilanka Perera, MD, PhD

Academic Editor

PLOS ONE

Additional Editor Comments (optional):

Reviewers' comments:

Reviewer's Responses to Questions

**Comments to the Author**

1. If the authors have adequately addressed your comments raised in a previous round of review and you feel that this manuscript is now acceptable for publication, you may indicate that here to bypass the “Comments to the Author” section, enter your conflict of interest statement in the “Confidential to Editor” section, and submit your "Accept" recommendation.

Reviewer #1: All comments have been addressed

2. Is the manuscript technically sound, and do the data support the conclusions?

Reviewer #1: Yes

3. Has the statistical analysis been performed appropriately and rigorously? 

Reviewer #1: Yes

4. Have the authors made all data underlying the findings in their manuscript fully available?

Reviewer #1: Yes

5. Is the manuscript presented in an intelligible fashion and written in standard English?

Reviewer #1: Yes

6. Review Comments to the Author

Reviewer #1: Thank you for addressing the comments. You have given clear explanations to all of them and amended the manuscript accordingly.

7. PLOS authors have the option to publish the peer review history of their article (what does this mean?). If published, this will include your full peer review and any attached files.

Reviewer #1: No

---

## [Editor Report · Acceptance letter]

27 Nov 2024

PONE-D-24-33301R1 

PLOS ONE

Dear Dr. Thanh , 

I'm pleased to inform you that your manuscript has been deemed suitable for publication in PLOS ONE. Congratulations! Your manuscript is now being handed over to our production team.

Kind regards, 

on behalf of

Dr. Nilanka Perera 

Academic Editor

PLOS ONE